# Evaluation of Change in Body Composition, including Phase Angle, in Post-Myocardial Infarction Patients Rehabilitated under the KOS-Zawał (MC-AMI) Programme

**DOI:** 10.3390/jcm13102784

**Published:** 2024-05-09

**Authors:** Aleksandra Ślązak, Iga Przybylska, Małgorzata Paprocka-Borowicz

**Affiliations:** 1Department of Physiotherapy, Division of Musculoskeletal Rehabilitation, Wroclaw Medical University, 50-368 Wroclaw, Poland; malgorzata.paprocka-borowicz@umw.edu.pl; 2Department of Physiotherapy in Internal Medicine, Wroclaw University of Health and Sport Sciences, 51-612 Wroclaw, Poland; iga.worwa@interia.pl

**Keywords:** phase angle, myocardial infarction, cardiac rehabilitation, electric impedance, MC-AMI, KOS-Zawał

## Abstract

**Background:** Ischaemic heart disease, including myocardial infarction, is one of the main causes leading to heart failure as a consequence of ischaemic myocardial damage. In recent years, survival in the acute phase of myocardial infarction has improved significantly, but the high mortality rate within 12 months of hospital discharge (reaching up to 9.8% in Poland) remains a challenge. Therefore, the KOS-Zawał (MC-AMI) comprehensive 12-month post-MI care programme was introduced in Poland in 2017. Aim: This study aimed to assess body composition (including, but not limited to, the phase angle, visceral fat, total body fat, redistribution between intracellular and extracellular fluid in the body, and metabolic age) using a bioelectrical impedance analysis (BIA) in post-MI patients before and after early post-MI rehabilitation who were participating in the KOS-Zawał (MC-AMI) programme. **Methods:** This study involved an examination (before rehabilitation) of 94 post-myocardial infarction patients who were referred to a cardiology appointment within 7–10 days of hospital discharge, during which a clinical assessment, electrocardiogram, and biochemical blood tests (complete blood count, CRP, and serum creatinine) were performed. For various reasons (death, qualification for device implantation, non-completion of rehabilitation, failure to attend a follow-up BIA), data from 55 patients who were examined twice (before and after rehabilitation) were used for the final analysis. Measurements were taken using a high-grade Tanita MC-780 BIA body composition analyser, which measured the resistance of tissues to a low-intensity electrical impulse (not perceptible to the subject). **Results:** Participation in rehabilitation as part of the KOS-Zawał (MC-AMI) programme was associated with a decrease in metabolic age in patients, with a reduction in visceral fat levels and levels of adipose tissue in the lower and upper limbs. Moreover, a clinically beneficial reduction in the ratio of extracellular water to total body water was also observed. These changes were statistically significant (*p* < 0.005). In contrast, there were no statistically significant differences in the change in phase angle values in the subjects before and after the 5-week post-infarction rehabilitation. **Conclusions:** Participation in early post-myocardial infarction rehabilitation as part of the KOS-Zawał (MC-AMI) programme (25 training sessions) is associated with significant improvements in body composition parameters, such as visceral adipose tissue, limb fat, and water redistribution, and, consequently, a reduction in metabolic age, despite no significant increase in phase angle values. It was hypothesised that the good baseline condition of the subjects might explain the lack of significant change in the phase angle over the short observation period. For further analysis, it would be worthwhile to increase the number of patients with baseline reduced phase angle values and monitor changes in this parameter throughout rehabilitation and the entire MC-AMI programme, because changes in the phase angle may also be influenced by other programme components such as dietary or psychological education. It is worth considering implementing a regular BIA assessment in patients in the programme as a motivating stimulus for diligent exercise and extending rehabilitation to be followed by telerehabilitation or hybrid telerehabilitation.

## 1. Introduction

Cardiovascular disease is the leading cause of morbidity and mortality in Europe [1]. The number of deaths from cardiovascular disease in ESC member countries far exceeds the number of deaths from cancer in both women and men. In 2020, 19.05 million CVD deaths were estimated worldwide, which is an increase of 18.71% compared to 2010 [1,2,3]. Ischaemic heart disease, including myocardial infarction, is one of the main causes of heart failure resulting from ischaemic damage to the heart muscle [4]. In Poland, as in many European countries, revascularisation in the acute phase of myocardial infarction is widely used. Revascularisation procedures are carried out in cardiac catheter laboratories that provide 24 h haemodynamic emergency services. In Poland, there is a network of 160 interventional cardiology centres, performing 735 percutaneous coronary interventions per million inhabitants [5,6]. These advanced methods of diagnosis and treatment have resulted in a significant reduction in early adverse events, including in-hospital deaths in the acute phase of myocardial infarction [7]. There has also been a reduction in 30-day mortality after myocardial infarction between 2007 and 2017 (the rate fell from 9.2% to 6.5% in European Union countries, including Poland) [8,9]. The persistently high post-MI mortality rate within 12 months of hospital discharge (reaching up to 9.8% in Poland) remains a clinical challenge [9,10].

A Cochrane systematic review of 85 randomised controlled trials shows clinical benefits in terms of reduced cardiovascular mortality, hospitalisation rates, and improved quality of life in patients with ischaemic heart disease who participated in cardiac rehabilitation. The aforementioned review also highlighted results showing that cardiac rehabilitation is financially cost-effective for healthcare systems compared to usual care [11]. Despite guideline recommendations on the importance of secondary prevention after myocardial infarction, cardiac rehabilitation programmes are still highly inaccessible and underutilised [12,13]. The above data were the reason for the introduction of the KOS-Zawał programme, i.e., the Managed Care after Acute Myocardial Infarction (MC-AMI) programme, in 2017 in Poland. The programme provides patients with timely access to rehabilitation (up to a maximum of 14 days after discharge from hospital following complete coronary revascularisation), as well as 12 months of outpatient cardiac care with biochemical monitoring, treadmill exercise testing, Holter ECG, and echocardiographic assessment (to identify patients eligible for ICD or CRT-D implantation for primary prevention of SCD). Cardiac rehabilitation is carried out by an interdisciplinary team enabling a holistic approach to the patient, and, consequently, dietary and psychological education is also present. The MC-AMI programme, which has been in operation for more than six years continuously, has undeniable clinical and economic benefits, which have been confirmed by numerous research studies showing short-term benefits as well as those from longer-term (12-month, 24-month) follow-up [4,6,9,14,15,16,17,18,19]. Those studies focused on a range of variables that affect observed outcomes, which, in turn, become drivers to further improve the system and introduce tools to further motivate patients.

Over the past few years, there has been a significant increase in interest in the use of bioelectrical impedance analysis (BIA) as an easy, accessible, and safe method of body composition analysis in various fields of medicine including cardiology. Bioelectrical impedance analysis is a technique used primarily to assess muscle mass, hydration, and the phase angle. With the phase angle, we can express the hydration state and assess the quality of cell membranes. A pulse of low-intensity current passes through the body and assesses the resistance of particular tissues (R resistance), providing data on water volume and hydration, and the reactance parameter (Xc) is used to assess the integrity of cell membranes [20]. This method also appears to be of interest in the context of post-MI patients, due to the possibility of assessing their body composition and observing changes in their bodies during the recovery period. It also appears to be useful because of the possibility of identifying additional factors burdening the patient or as an element of motivation to work intensively on lifestyle changes.

The above theses are based on studies conducted on various groups of subjects (including active young adults [21], physically active elderly people [22,23], professional soldiers [24], or people with cardiovascular disease [3,25,26,27]) using bioelectrical impedance. There have been articles in the literature showing that bioelectrical impedance parameters, including the phase angle, can be a good marker of sarcopenia, malnutrition, and cachexia in people with cardiovascular disease, all of which are factors for poorer prognosis [28,29].

### Objectives

This study aimed to assess body composition (including, but not limited to, the phase angle, visceral fat, total body fat, distribution of intracellular and extracellular fluid in the body, and metabolic age) using BIA in post-MI patients before and after early post-MI rehabilitation who were participating in the KOS-Zawał (MC-AMI) programme.

## 2. Materials and Methods

This study was approved by the Bioethics Committee of the Wroclaw Medical University (approval number KB 275/2021) and conducted in accordance with Good Clinical Practice Guidelines and the Declaration of Helsinki. All project participants were informed about the purpose and the method of the study and gave their written consent to participate and to have their personal data processed.

### 2.1. Study Participants and Study Setting

The project was conducted as a retrospective analysis of prospectively collected data of 94 patients (examined before commencing rehabilitation) who were referred within 7–10 days of hospital discharge for a cardiology appointment during which clinical assessment, electrocardiogram, biochemical blood tests (complete blood count, CRP, serum creatinine) were performed. The assumption of the KOS-Zawał programme (MC-AMI) was that doctors who discharged a patient from the cardiac ward after a myocardial infarction qualified the patient to a rehabilitation centre (a day centre or a hospital-based rehabilitation centre in the case of patients with multiple additional clinical burdens).

In order to qualify a patient after myocardial infarction for hospital-based rehabilitation, it is necessary to have a coexisting diagnosis (Table 1 [30]) or 3rd grade of disability according to the Modified Rankin Scale (Table 2 [30]).

Therefore, patients who came to our day centre rehabilitation were in a better clinical condition, circulatory and respiratory compensated.

The patient enrolment pattern in our study was consistent with cardiac rehabilitation guidelines. The first stage was conducted in hospital and lasted until the patient was discharged. It was started as early as possible, immediately after the patient’s condition stabilised in the conditions of an intensive supervision room.

The type of model in the first stage of rehabilitation in patients after myocardial infarction was carried out depending on clinical diagnosis and left ventricular function.

Model A1 (4–7 days) included patients with non-ST elevation myocardial infarction (NSTEMI) and ST-elevation myocardial infarction (STEMI) without significant impairment of left ventricle.

Model A2 (7–10 days) included patients with ST elevation myocardial infarction with significant impairment of left ventricular function.

Model B (>10 days) included patients with complicated myocardial infarction (significantly impaired left ventricle, persistent symptoms of heart failure, delayed infarction evolution, dyskinesia in echocardiographic examination).

In the first stage of rehabilitation, the principle of gradual mobilisation of the patient applies. The rate of mobilisation is determined by the presence of risk factors for complications (Table 3 [30]).

Second stage of early cardiac rehabilitation should last 4–12 weeks. According to the guidelines of the KOS-Zawał programme, it should be started within 14 days after discharge from the end of the full-cycle revascularisation.

Patients had their second BIA examination performed within 7 days of completing rehabilitation. They could not be examined on the day of the exercise—the condition for a reliably conducted examination is, among others, prohibition of intense exercise within 12 h before the measurement. Therefore, patients had to come for the BIA test on a special date.

A total of 39 patients were not examined twice, of whom 2 died, 1 person was qualified for ICD implantation, 3 patients underwent repeated coronary angioplasty (rehabilitation was suspended), and 7 people did not return for another examination due to family reasons (8 people—follow-up examination coincided with the holiday season; 18 people—they did not fit into assumed 7 days due to the need for periodic body composition analyser validation). Most of these patients had the second BIA examination at another time because they wanted to compare the results, but these results were not included in the study. For these reasons, data from 55 patients who were examined twice (before and after rehabilitation) were ultimately used for the final analysis.

Ultimately, the final examination included post-revascularisation patients who were discharged from cardiac wards with a recommendation for cardiac rehabilitation under the MC-AMI programme. All patients were Caucasian and were treated for mobility issues at the Pro Corde Day Rehabilitation Centre in Wrocław from 1 April to 30 December 2021. Patients who were qualified to a hospital-based rehabilitation centre were not included in the study. Patients were qualified for the project by an interdisciplinary team consisting of a cardiologist, a medical rehabilitation specialist, and a physiotherapist. The selection of patients to participate in the study was purposive.

If a patient required a further stage of revascularisation, he or she was referred back to hospital at an agreed date. In the next stage, patients were referred to a rehabilitation centre (a day centre or a hospital-based rehabilitation centre in the case of patients with multiple additional clinical burdens, e.g., insulin-dependent diabetes mellitus, heart failure with significantly reduced left ventricular ejection fraction) where, within 14 days of hospital discharge, they started a 5-week cycle of physical rehabilitation. Six weeks after hospitalisation due to myocardial infarction, patients had a follow-up echocardiogram to assess left ventricular ejection fraction and possible qualification for implantation of a cardioverter-defibrillator or cardiac resynchronisation device. Patients remained under cardiological care for 12 months after their MI, which involved four mandatory medical appointments (including a follow-up visit just before the end of participation in the programme). During the medical appointments, follow-up laboratory tests and Holter ECG tests were requested. The programme also provided dietary and psychological counselling. At the beginning, as well as at the end of the programme, each patient participated in an exercise test.

All participants were required to meet the following eligibility criteria: age ≥ 18 years, status post myocardial infarction (within 14 days of hospital discharge), participation in early rehabilitation as part of the MC-AMI programme, and adherence to the following: no eating or drinking within 3 h before measurement, no intense exercise within 12 h before measurement, no alcohol within 12 h before measurement. If possible, they adhered to the following: urination before measurement; no large amounts of food or drink on the day preceding the measurement; in the case of women, no measurement during menstruation; and signing an informed consent to participate in the study.

Excluded from the study were patients who did not consent to participate in the study or had electronic implants such as pacemakers, cardioverter-defibrillators and cardiac resynchronisation devices, as well as pregnant women.

Measurements were taken using a high-grade Tanita MC-780 body composition analyser, which measured the resistance of tissues to a low-intensity electrical impulse (not perceptible to the subject). Measurements were taken twice (before and after rehabilitation). Before analysis, anthropometric measurements (height), date of birth, gender, and race were entered. The patient was examined in a standing position—standing with bare feet on the scale, holding the electrodes in both hands while keeping their hands straight at the elbows, at a 45-degree angle away from the torso.

Patients who qualified for post-infarction rehabilitation also performed an exercise test on a treadmill according to the Bruce modified protocol. Based on the result of the exercise test, the patient’s optimal heart rate, which should be achieved during exercise, was determined. The so-called Karvonen formula was used for calculations: target heart rate = (maximum heart rate − resting heart rate) × factor + resting heart rate. For patients with a recent myocardial infarction, a factor of 0.6 or 0.8 was used according to age and the number of METs obtained during the exercise test (MET—metabolic equivalent for estimating the intensity of physical activity).

Patients were rehabilitated according to the following scheme. Each unit of rehabilitation treatment lasted 45–60 min. The rehabilitation training period lasted 25 days. For patient safety reasons, blood pressure and heart rate were measured before each kinesitherapy unit. Rehabilitation treatment was always tailored to each individual, taking into account the patient’s possibilities. Before kinesitherapy, a chest pulse monitor was attached (in the case of training on a cycloergometer, a pulse monitor with a recording of the 1st ECG lead) and the current heart rate level of each patient was displayed on a monitor during exercise, with an alarm signal when the desired heart rate was exceeded and verbal encouragement from the physiotherapist to intensify training when the heart rate was too low.

Training sessions were held in 3 rooms (in alternating mode, a different room each day): gymnasium with general aerobic exercises (45–60 min), station training (7–8 min of exercises, then a 5 min break, with treadmill, elliptical bike, multi gym, rowing machine, and stepper) (45–60 min), and interval training on cycloergometers (45 min; the training was shorter because there was no break).

### 2.2. Statistical Analysis

Statistical analysis was performed using Statistica 13.1 software (TIBCO, Inc., 3307 Hillview Ave, Palo Alto, CA 94304, USA). Arithmetic means, medians, standard deviations, quartiles, and range of variation (extreme values) were calculated for measurable variables. Prevalence (per cent) was calculated for qualitative variables. All analysed quantitative variables were verified with the Shapiro–Wilk test used for the determination of the distribution type. Within-group comparisons between pre- and post-rehabilitation scores were made using the Wilcoxon test. The level of α = 0.05 was used for all comparisons.

## 3. Results

Ultimately, 55 myocardial infarction survivors enrolled in the MC-AMI programme participated in the study. The baseline characteristics of the sample group are shown in Table 4. The characteristics of clinical variables in our group of patients are shown in Table 5.

Women comprised 25% (*n* = 14) of the study participants, while men comprised 75% (*n* = 41). The mean age was 61.0 years (min–max: 34.0–88.0 years; SD = 11.7 years), the mean body height was 171.9 centimetres (min–max: 149.0–192.0 cm; SD = 9.3 cm), the mean body weight was 83.0 kilograms (min–max: 54.6–137.4 kg: SD = 16.7 kg), and the mean BMI was 27.9 (min–max: 20.1–37.6; SD = 4.3) (Table 4).

Patients had two measurements on a Tanita body composition analyser (before and after rehabilitation). Participation in rehabilitation as part of the MC-AMI programme was associated with a statistically significant decrease in metabolic age in patients—the mean result before rehabilitation was 59.0 years (min–max: 37.0–88.0 years; SD = 11.8 years) and was 2.7 years higher than that obtained after rehabilitation, with a mean of 56.3 years (min–max: 37.0–88.0 years; SD = 11.2 years). The results were statistically significantly different (*p* < 0.05) (Table 6). There was a reduction in the level of visceral fat—the mean result before rehabilitation was 38.7% (min–max: 5.0–79.8%; SD = 17.8%) and was 2.3% higher than that obtained after rehabilitation, with a mean of 36.4% (min–max: 5.0–79.8%; SD = 17.8%), the scores being statistically significantly different (*p* < 0.05) (Table 7). There was a reduction in the level of lower limb body fat—the mean result before rehabilitation was 7.71 kg (min–max: 2.70–15.30 kg; SD = 3.24 kg) and was 0.40 kg higher than the post-rehabilitation scores, with a mean of 7.11 kg (min–max: 2.50–14.60 kg; SD = 3.06 kg); this result was statistically significantly different (*p* < 0.05) (Table 8). There was a reduction in upper limb body fat—the mean result before rehabilitation was 2.41 kg (min–max: 1.00–5.60 kg; SD = 1.05 kg) and was 0.11 kg higher than the scores obtained after rehabilitation, with a mean of 2.30 kg (min–max: 1.00–5.40 kg; SD = 0.99 kg); this result was statistically significantly different (*p* < 0.05) (Table 9).

What is more, a clinically beneficial reduction in the ratio of extracellular water to total body water was also observed—the mean result before rehabilitation was 0.444 (min–max: 0.403–0.503; SD = 0.024) and was 0.005 higher than that after rehabilitation, with a mean of 0.439 (min–max: 0.402–0.487; SD = 0.024); (*p* < 0.05) (Table 10). However, there were no statistically significant differences in the change in phase angle values in the subjects before and after the 5-week post-myocardial infarction rehabilitation (*p* > 0.05) (Table 11).

## 4. Discussion

Considering the group of 55 patients in the present study rehabilitated over a 5-week exercise cycle, there were no statistically significant changes in phase angle values. This group was heterogeneous in terms of age and baseline exercise capacity. Similar findings were reported by Lira et al., who examined a group of young, active individuals (also observed after 5-week exercise periods) and obtained no significant changes in phase angle values [21]. In contrast, Campa et al., in their study involving elderly patients, showed a positive effect of resistance training on improving phase angle values [22]. It was therefore hypothesised that a good baseline condition of the subjects might explain the absence of a significant change in the phase angle over the short observation period. For further analyses, it would be worthwhile to increase the number of patients with baseline reduced phase angle values and monitor changes in this parameter during the rehabilitation and the entire MC-AMI programme.

However, even during such a short follow-up of post-MI patients as the 5-week exercise cycle (25 training sessions), significant improvements can be observed in other body composition parameters, such as visceral fat, limb fat, and water redistribution, and can consequently lead to a reduction in metabolic age.

Also evident in the present study was the motivating effect of diligent exercise in patients assessed with a body composition analyser before commencing rehabilitation. Given the growing interest in bioimpedance in medicine (including cardiology), in the future it would be worth considering including the assessment of these parameters in post-MI patients as a permanent part of the programme.

Based on these facts, it can be assumed that the inclusion of body composition assessment using BIA in patients participating in the said programme could have synergistic benefits in long-term follow-up. Further studies are needed to confirm these preliminary findings.

Despite the well-known benefits of cardiac rehabilitation, there is still a lack of sufficient methods to motivate patients to participate in this form of therapy, especially for the female group [11,15].

Studies show that obesity, and the associated increased adipose tissue, have a strongly pro-inflammatory effect. Garcia-Rubira et al. showed that individuals with a metabolic age (assessed using BIA) higher than their actual age had higher levels of body fat and higher levels of the pro-inflammatory Interleukin 1b (Il-1b), and those parameters were positively correlated with the FRS scale (Framingham Risk Score, including factors associated with a distant 10-year increased risk of developing coronary heart disease) [31].

It can be speculated that improvements in body composition and the phase angle are influenced by the length and regularity of the training provided. Such positive changes were demonstrated after young men participated in a 6-month training cycle according to strict military guidelines [24]. Furthermore, Tuesta et al. assessed the effect of cardiac rehabilitation in patients who participated in different lengths of exercise cycles (from 24 to 36 training sessions). They attempted to determine the optimal length of exercise sessions and concluded that the recommended minimum number of exercise sessions should be 24, and preferably 36 [32,33]. Considering the above, it would be reasonable to ask whether the changes in body composition (including the phase angle) assessed using BIA in patients from the MC-AMI programme would have been even more satisfying if the number of training sessions was increased by extending the rehabilitation to be followed with telerehabilitation and hybrid telerehabilitation. The effectiveness of this method (reduction in the risk of death compared to patients not participating in rehabilitation), its safety, and its acceptance by patients were confirmed in a study by Orzechowski et al. [9].

Wita et al. [25], in their study conducted on a large group of patients (more than 10,000 patients from 16 centres treating acute coronary syndromes), found a 38% reduction in all-cause mortality compared to the control group (after propensity matching) at 12-month follow-up. This effect persisted even after the programme ended. After analysing the different stages of the MC-AMI programme, they found that not only rehabilitation, but also further outpatient care with education, had an impact on the reduction in total mortality.

### Limitations

There are some limitations to this study that need to be taken into account. This study was conducted in a single centre, only on Caucasian patients, and the sample group was heterogeneous in terms of age and it was small. Furthermore, the exact number of absences from exercise during the 25 days of rehabilitation was not taken into account.

Another limitation is the absence of a control group.

This study involved the assessment of bioimpedance parameters before and after rehabilitation (approximately 2–3 months after commencing participation in the programme). Despite the promising results after such a short follow-up, it would be worthwhile to assess how body composition parameters change after a 12-month programme, as the final results (including the value of the phase angle) might still be influenced by other components of the programme such as drug treatment, dietary and psychological education, and continued physical activity at home. Further studies are needed to confirm these preliminary findings.

## 5. Conclusions

Participation in early cardiac rehabilitation as part of the KOS-Zawał (MC-AMI) programme significantly reduces visceral adipose tissue and upper and lower limb adipose tissue, in addition to having a positive effect on body water redistribution (ratio of extracellular to total body water). These changes in body composition parameters have the effect of significantly reducing the metabolic age in post-myocardial infarction patients. In contrast, there was no significant increase in phase angle values after participating in 25 days of post-infarction rehabilitation.

## Figures and Tables

**Table 1 jcm-13-02784-t001:** Comorbidities that enable qualifying a patient after a myocardial infarction to hospital-based rehabilitation centre ([30] with own modification).

1. Previous cardiac surgery
2. Cancer
3. Heart failure (EF ≤ 35% or EF > 35% for patients in NYHA class III)
4. Complicated course of revascularisation or surgical treatment of acute myocardial infarction
5. Comorbidities requiring increased care such as the following:
- COPD with acute lower respiratory tract infection;
- COPD with severe symptoms or a high risk of exacerbations;
- Insulin-dependent diabetes;
- End-stage renal disease (GFR < 15 mL/min/1.73 m^2^ or dialysis treatment.

**Table 2 jcm-13-02784-t002:** Modified Rankin Scale (MRS) ([30] with own modification).

Score Description
Grade 0 No symptoms at all
Grade 1 No significant disability despite symptoms; able to carry out all usual duties and activities
Grade 2 Slight disability; unable to carry out all previous activities, but able to look after own affairs without assistance
Grade 3 Moderate disability; requiring some help, but able to walk without assistance
Grade 4 Moderately severe disability; unable to walk without assistance and unable to attend to own bodily needs without assistance
Grade 5 Severe disability; bedridden, incontinent and requiring constant nursing care and attention

**Table 3 jcm-13-02784-t003:** Risk factors for complications influencing the choice of the rehabilitation model ([30] with own modification).

- Low ejection fraction, symptoms of congestive heart failure
- Extensive segmental disturbances of left ventricular contractility, presence of dyskinesia or left ventricular aneurysm
- Presence of a thrombus in the left ventricle
- Previous cardiac arrest or serious ventricular arrest heart rhythm disturbances in the acute period of the disease
- Cardiogenic shock, pulmonary edema, thromboembolic episode in the acute period of the disease
- Changes in coronary vessels: trunk stenosis left coronary artery, trunk equivalent, disease three-vessel coronary artery with significant changes in the proximal sections of the arteries
- Coronary ailments with little physical effort

**Table 4 jcm-13-02784-t004:** The baseline characteristics of the sample group.

Group *n* = 55								
Variable		$\bar{x}$	Mdn	Min	Max	Q1	Q3	SD
Age [years]		61.0	61.0	34.0	88.0	53.0	71.0	11.7
Height [cm]		171.9	172.0	149.0	192.0	168.0	178.0	9.3
Weight [kg]		83.0	82.0	54.6	137.4	68.8	94.1	16.7
BMI		27.9	27.8	20.1	37.6	24.6	30.7	4.3
Variable	Category of variable	*n*			%			
Gender	Female	14			25.5			
	Male	41			75.5			

$\bar{x}$—mean; Mdn—median; Q1—first quartile; Q3—third quartile; Min—minimum value; Max—maximum value; SD—standard deviation; *n*—number of subjects; %—percentage of subjects.

**Table 5 jcm-13-02784-t005:** The characteristics of clinical variables in our group of patients.

Variables	Patients with KOS (*n* = 55)
Age, years, mean (SD)	61.0 (11.7)
Sex	
Male n%	75% (41)
Female n%	25% (14)
Kidney failure n%	7.2% (4)
LVEF (%) < 45%, n%	9% (5)
Hypertension n%	76% (42)
Stroke in history n%	1.8% (1)
Diabetes (oral therapy) n%	20% (11)
Cancer n%	5.4% (3)
COPD n%	5.4% (3)

Descriptive data are presented as mean (SD) or number (%). Abbreviations: COPD, chronic obstructive pulmonary disease; KOS-Zawał, comprehensive coordinated care after myocardial infarction; LVEF left ventricular ejection fraction.

**Table 6 jcm-13-02784-t006:** Comparison of changes in metabolic age scores before and after rehabilitation.

Group *n* = 55								
Variable	Measurement	$\bar{x}$	Mdn	Min	Max	Q1	Q3	SD
METAAGE [years]	Before	59.0	58.0	37.0	88.0	50.0	68.0	11.8
	After	56.3	56.0	37.0	88.0	48.0	64.0	11.2
*p*-value *		<0.001						

$\bar{x}$—mean; Mdn—median; Min—minimum value; Max—maximum value; Q1—lower quartile; Q3—upper quartile; SD—standard deviation; * Wilcoxon signed-rank test. METAAGE—metabolic age.

**Table 7 jcm-13-02784-t007:** Comparison of changes in visceral fat scores before and after rehabilitation.

Group *n* = 55								
Variable	Measurement	$\bar{x}$	Mdn	Min	Max	Q1	Q3	SD
VISCFAT [%]	Before	38.7	39.6	5.0	79.8	27.6	50.1	17.8
	After	36.4	39.6	5.0	79.8	21.4	45.1	17.8
*p*-value *		0.009						

$\bar{x}$—mean; Mdn—median; Min—minimum value; Max—maximum value; Q1—lower quartile; Q3—upper quartile; SD—standard deviation; * Wilcoxon signed-rank test. VISCFAT—visceral fat.

**Table 8 jcm-13-02784-t008:** Comparison of changes in lower limb fat mass scores before and after rehabilitation.

Group *n* = 55								
Variable	Measurement	$\bar{x}$	Mdn	Min	Max	Q1	Q3	SD
RLFATM + LLFATM [kg].	Before	7.71	7.20	2.70	15.30	5.20	9.70	3.24
	After	7.11	6.80	2.50	14.60	5.00	8.10	3.06
*p*-value *		<0.001						

$\bar{x}$—mean; Mdn—median; Min—minimum value; Max—maximum value; Q1—lower quartile; Q3—upper quartile; SD—standard deviation; * Wilcoxon signed-rank test. RLFATM—right leg fat mass; LLFATM—left leg fat mass.

**Table 9 jcm-13-02784-t009:** Comparison of changes in upper limb fat mass scores before and after rehabilitation.

Group *n* = 55								
Variable	Measurement	$\bar{x}$	Mdn	Min	Max	Q1	Q3	SD
RAFATM + LAFATM [kg].	Before	2.41	2.10	1.00	5.60	1.60	3.20	1.05
	After	2.30	2.00	1.00	5.40	1.60	2.80	0.99
*p*-value *		0.004						

$\bar{x}$—mean; Mdn—median; Min—minimum value; Max—maximum value; Q1—lower quartile; Q3—upper quartile; SD—standard deviation; * Wilcoxon signed-rank test. RAFATM—right arm fat mass; LAFATM—left arm fat mass.

**Table 10 jcm-13-02784-t010:** Comparison of changes in extracellular to total body water ratio scores before and after rehabilitation.

Group *n* = 55								
Variable	Measurement	$\bar{x}$	Mdn	Min	Max	Q1	Q3	SD
ECW/TBW	Before	0.444	0.442	0.403	0.503	0.426	0.459	0.024
	After	0.439	0.438	0.402	0.487	0.419	0.452	0.024
*p*-value *		<0.001						

$\bar{x}$—mean; Mdn—median; Min—minimum value; Max—maximum value; Q1—lower quartile; Q3—upper quartile; SD—standard deviation; * Wilcoxon signed-rank test. ECW—extracellular water; TBW—total body water.

**Table 11 jcm-13-02784-t011:** Comparison of changes in phase angle results before and after rehabilitation.

Group *n* = 55								
Variable	Measurement	$\bar{x}$	Mdn	Min	Max	Q1	Q3	SD
Phase Angle [°]	Before	5.51	5.72	3.64	6.78	4.94	6.14	0.84
	After	5.53	5.65	3.82	7.10	4.81	6.16	0.83
*p*-value *		0.46						

$\bar{x}$—mean; Mdn—median; Min—minimum value; Max—maximum value; Q1—lower quartile; Q3—upper quartile; SD—standard deviation; * Wilcoxon signed-rank test.

## Data Availability

The authors confirm that all data underlying the findings described in this manuscript are fully available to all interested researchers upon request.

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
