# Peer review of "Evaluation of Change in Body Composition, including Phase Angle, in Post-Myocardial Infarction Patients Rehabilitated under the KOS-Zawał (MC-AMI) Programme"

_jcm, 2024, doi:10.3390/jcm13102784_

Round 1

Reviewer 1 Report

Comments and Suggestions for Authors

The presentation is interesting, but apart from the mentioned shortcomings that the authors also presented, I am of the opinion that there is also a lack of other data that would be important for early rehabilitation.

Maybe it would be better to make groups according to DM and non-DM diseases or if there are other assessments of hypervolemia! Clinical examination or echo of the heart and inferior vena cava. Pleural effusions zas or no! Laboratory data Hct or RDW are also missing ... the work seems incomplete!

Author Response

Dear Reviewer, 

thank you very much for taking the time to review this manuscript. Your suggestions were very precious and made me think more about the study. I'm sure some of them I will try to use in another sudies in the future because the issue of the cardiac rehabilitation and bioelectric impedance are exciting topics for me. In my opinion it is a lot to do to improve effectiveness of treatment the patients after myocardial infarction.

I hope you will be so kind to look at my response below:

Comments:

The presentation is interesting, but apart from the mentioned shortcomings that the authors also presented, I am of the opinion that there is also a lack of other data that would be important for early rehabilitation. Maybe it would be better to make groups according to DM and non-DM diseases or if there are other assessments of hypervolemia! Clinical examination or echo of the heart and inferior vena cava. Pleural effusions zas or no! Laboratory data Hct or RDW are also missing ... the work seems incomplete!

  1. I added the description of clinical characteristics of the patients who were not included in the study (line 129-150, 169-170, Fig. 1 and Fig.2). Each patient who was qualified to the day rehabilitation centre (Pro Corde) and met the inclusion criteria was offered to participate in the study. (I added the new citation to the bibliography, No 33) Patients who were qualified to a hospital- based rehabilitation centre were not included in the study. 

    Therefore, patients who came to our day centre rehabilitation were in a better clinical condition, circulatory and respiratory compensated.

I would like to thank you for any suggestions and I hope I managed to answer your comments.

Reviewer 2 Report

Comments and Suggestions for Authors

I am grateful to the editor for the opportunity to review the manuscript by Aleksandra Maria ÅšlÄ…zak et al “Evaluation of change in body composition, including phase angle, in post-myocardial infarction patients rehabilitated under the KOS-zawaÅ‚ (MC-AMI) program”. In this article, the authors provide data on the effect of a 5-week cardiac rehabilitation program in patients after myocardial infarction on bioelectrical impedance analysis indicators, in particular on phase angle. In fact, the analyzed cohort of patients has already shown the beneficial effect of this rehabilitation program on a wide range of clinical indicators, as indicated in this article. The difference between this manuscript is that it analyzes the dynamics of bioelectrical impedance analysis indicators during rehabilitation, and above all the phase angle. This indicator is a very fashionable integral indicator of the state of metabolic health, used both in healthy individuals and in patients.

During the review, I had comments and questions that I would like answers to from the authors:

1. The ABSTRACT contains links to publications, this is unnecessary and needs to be corrected.

2. The submitted manuscript does not indicate the authors or their affiliations.

3. The authors did not provide a flow chart for the inclusion of patients in the study.

4. The article does not contain clinical characteristics of the patients, which does not allow us to understand for which category of patients this rehabilitation program was used.

5. There is a large number of patients who dropped out; a more detailed description of the reasons for dropping out from the study is required.

6. The format of tables 2-7 requires correction - it is best to combine them into one table. Indicators in the tables (VISCFAT, RLFATM, LLFATM, etc.) require explanation in the notes.

7. Figures 1-6 do not provide additional information (it is all presented in tables), so they must be removed from the text of the manuscript.

8. The Discussion section should begin with the data obtained, and not with literary references.

9. References 30 and 32 in the bibliography are incomplete and require correction.

10. The section Limitations of the study requires additions; in particular, it is necessary to indicate the absence of a control group. In this case, there is no certainty that the observed dynamics of bioelectrical impedance analysis indicators arose precisely as a result of the rehabilitation program, and not as a result of the natural course of the post-infarction period. It is also necessary to indicate the high percentage of dropouts from the study.

11. The authors in the Discussion section indicate that “Also evident in the present study is the motivating effect of diligent exercise in patients assessed with a body composition analyzer before commencing rehabilitation” (lines 311-312). However, the large percentage of those who dropped out of the study does not fit well with this statement - it is difficult to agree with the increased motivation of the participants.

Comments on the Quality of English Language

No comments

Author Response

Dear Reviewer, 

thank you very much for taking the time to review this manuscript. Your suggestions were very precious and made me think more about the study. I'm sure some of them we will try to use in another sudies in the future because the issue of the cardiac rehabilitation and bioelectric impedance are exciting topics for me. In my opinion it is a lot to do to improve effectiveness of treatment the patients after myocardial infarction.

I hope you will be so kind to look at my response below:

  1. The ABSTRACT contains links to publications, this is unnecessary and needs to be corrected. - I removed links to publication
  2. The submitted manuscript does not indicate the authors or their affiliations. -I added affiliation
  3. The authors did not provide a flow chart for the inclusion of patients in the study. - I added the description of clinical characteristics of the patients who were not included in the study (line 129-150, 169-170, Fig. 1 and Fig.2). Each patient who was qualified to the day rehabilitation centre (Pro Corde) and met the inclusion criteria was offered to participate in the study. (I added the new citation to the bibliography, No 33) Patients who were qualified to a hospital- based rehabilitation centre were not included in the study.

    4. The article does not contain clinical characteristics of the patients, which does not allow us to understand for which category of patients this rehabilitation program was used. - I added the description of clinical characteristics of the patients who were not included in the study . (lines  129-150, 169-170, Fig. 1 and Fig.2)

    5. There is a large number of patients who dropped out; a more detailed description of the reasons for dropping out from the study is required. -

    I added more detailed description of the reasons of dropping out from the study (lines 151-164)

    6. The format of tables 2-7 requires correction - it is best to combine them into one table. Indicators in the tables (VISCFAT, RLFATM, LLFATM, etc.) require explanation in the notes. - I added explanations of the indicators in the tables 2-7 . I left the tables separate for better clarity

    7. Figures 1-6 do not provide additional information (it is all presented in tables), so they must be removed from the text of the manuscript. - These figures removed. I added the new figures No 1 and 2 (see above; my response for the Comments 3 and 4 )

    8. The Discussion section should begin with the data obtained, and not with literary references. - I changed the order of information posted in the Discussion so that at the beginning there would be mainly the data we obtained. 

    9. References 30 and 32 in the bibliography are incomplete and require correction. - I changed citation style

    10. The section Limitations of the study requires additions; in particular, it is necessary to indicate the absence of a control group. In this case, there is no certainty that the observed dynamics of bioelectrical impedance analysis indicators arose precisely as a result of the rehabilitation program, and not as a result of the natural course of the post-infarction period. It is also necessary to indicate the high percentage of dropouts from the study. - I added the information about the absence of the control group (line 366)

    11. The authors in the Discussion section indicate that “Also evident in the present study is the motivating effect of diligent exercise in patients assessed with a body composition analyzer before commencing rehabilitation” (lines 311-312). However, the large percentage of those who dropped out of the study does not fit well with this statement - it is difficult to agree with the increased motivation of the participants. - The large percentage of those who dropped out of the study was not related to lack of motivation of the participants (lines 150-164).

    I would like to thank you for any suggestions and I hope I managed to answer your comments.

Round 2

Reviewer 1 Report

Comments and Suggestions for Authors

/

Author Response

Dear Reviewer, 

thank you very much for taking the time to review this manuscript and for all your suggestions.

Kind regards,

Reviewer 2 Report

Comments and Suggestions for Authors

The authors made corrections to the text of the manuscript and answered my questions. However, these answers did not completely satisfy me.

3. The authors did not provide a flow chart for inclusion of patients.

4. The authors did not provide data on the clinical characteristics of the patients. Table 1 presents only the inclusion/exclusion criteria for patients

Comments on the Quality of English Language

No comments

Author Response

Dear Reviewer, 

thank you very much for taking the time to review this manuscript. Your suggestions were very precious and made me think more about the study and I tried to collect the missing data.

I hope you will be so kind to look at my response below:

3. The authors did not provide a flow chart for inclusion of patients.-  I added a scheme for inclusion of patients (Figure 3)-marked in yellow

4. The authors did not provide data on the clinical characteristics of the patients. - I added the description of clinical characteristics of the patients (table 8)- marked in yellow

I would like to thank you for any suggestions and I hope I managed to answer your comments.

Round 3

Reviewer 1 Report

Comments and Suggestions for Authors

Nop

Reviewer 2 Report

Comments and Suggestions for Authors

The authors answered my questions again and made additional corrections to the text of the manuscript. This made it possible to improve the article, I have no other comments.

Comments on the Quality of English Language

No comments